# ‘I Felt Welcomed in Like They’re a Little Family in There, I Felt Like I Was Joining a Team or Something’: Vape Shop Customers’ Experiences of E-Cigarette Use, Vape Shops and the Vaping Community

**DOI:** 10.3390/ijerph16132341

**Published:** 2019-07-02

**Authors:** Tessa Langley, Rebecca Bell-Williams, Julie Pattinson, John Britton, Manpreet Bains

**Affiliations:** UK Centre for Tobacco and Alcohol Studies, Division of Epidemiology and Public Health, University of Nottingham, Nottingham NG7 2RD, UK

**Keywords:** e-cigarettes, smoking cessation, vaping, tobacco control, harm reduction, vape shops

## Abstract

*Background*: Specialist electronic cigarette (e-cigarette) shops, known as vape shops, provide access to a less harmful alternative to smoking. This study aimed to understand customers’ experiences of vaping and vape shops, and the extent to which smoking cessation advice is and should be provided in these shops. *Methods*: We conducted telephone interviews with 22 customers recruited in vape shops in the East Midlands region of England. Interviews explored participants’ smoking histories, reasons for using e-cigarettes, the role of vape shops in their e-cigarette use, and whether smoking cessation was discussed in vape shops. Interviews were analysed following framework approach principles. *Results*: Most respondents regarded e-cigarettes as a quitting tool and reported very positive experiences of vaping. Vape shops were central to participants’ positive experiences, in that they provided access to a wide variety of high-quality products and reliable product information and advice. The shop staff engendered a sense of loyalty in customers which, together with the community of other vapers, created a network that helped to support e-cigarette use. Vape shops were not regarded as a setting in which cessation advice was generally provided but were acknowledged as potentially appropriate places to provide quitting support. *Conclusions*: Vape shops have the potential to play an important role in tobacco harm reduction, which could be increased if their service model were to extend to help smokers to quit.

## 1. Introduction

Electronic cigarettes (e-cigarettes) are the most commonly used smoking cessation aid in England and have been endorsed by a range of organisations [1,2,3,4]. Over 3 million adults in Britain currently use e-cigarettes; most are ex-smokers who use e-cigarettes to help them quit or remain abstinent [5]. In the UK, e-cigarettes are available on general sale, with specialist ‘vape shops’ as the most popular source of products [1]. There are currently in the region of 2000 vape shops in the UK [6].

Vape shops provide easy access to a less harmful alternative to smoking [2] and are able to respond to market developments quickly. Their customers are likely to be interested in quitting smoking or cutting down, and staff interactions with customers can provide opportunities to promote and maintain smoking cessation [7]. This makes it particularly important to understand the role of vape shops in e-cigarette use and supporting quitters [7]. However, relatively little is known about these shops’ customers, or the information vape shops provide, particularly outside of the USA, where the majority of existing studies have been conducted [8,9,10,11,12,13,14,15]. A recent quantitative study of vape shops in the East Midlands region of England found that most vape shops customers were ex-smokers who were using e-cigarettes to quit or stay quit [16]. The study also suggested that the majority of advice provided by vape shop staff was product—rather than cessation—focussed, but provided only limited insight into the interactions between staff and customers, or customers’ experiences of vaping and vape shops. A qualitative study investigating the role of the vape shop environment in supporting smoking abstinence in East Anglia, Kent and London found that traditional smoking cessation is not perceived as the main role of vape shops by either vapers or vape shop staff, but it is unclear whether its findings are representative of other settings [7]. We now report a qualitative study of vape shop customers in the East Midlands, which aimed to understand vape shop customers’ experiences of vaping and vape shops and the need and potential for offering smoking cessation advice in a vape shop setting.

## 2. Materials and Methods

### 2.1. Sampling and Recruitment

Participants were recruited from respondents to a survey of staff and customers from 41 vape shops across the East Midlands, UK [16], in which participants were asked to indicate whether they would be willing to be contacted by the research team to take part in a qualitative interview. Written consent to take part in a qualitative interview was obtained from those who were willing to be contacted. A researcher (JP) then attempted to contact consenting individuals to arrange a convenient time to complete a telephone interview. Participants were recruited in sequence of response and recruitment ceased when the initial readings of transcripts indicated data had reached saturation, after a total of 22 interviews.

### 2.2. Data Collection

Interviews were conducted in April and May 2016 via telephone, and consent reconfirmed verbally. A semi-structured guide was used to facilitate discussions and explore participants’ smoking histories, reasons for using e-cigarettes, general experiences of vape shops and product choice, and the extent to which smoking cessation was discussed in vape shop settings. Interviews were audio-recorded and lasted an average of 20 min (range 13–34). The interviews were transcribed verbatim and personal identifiable information was anonymised.

### 2.3. Analysis

Interviews were analysed following framework approach principles [17,18], with theme identification inductive in nature. To facilitate familiarisation with the data, RB read the transcripts several times, identifying initial codes, themes and sub-themes. To minimise researcher bias and enhance validity, the analysis process employed investigator triangulation, whereby MB and TL independently coded 10 data-rich transcripts to identify initial themes that were then discussed between the researchers to reach consensus on the thematic framework. This framework was applied and refined following analysis of the remaining transcripts. Using NVivo 10 (QSR International, Melbourne, Australia), data were then indexed according to the final framework and charted into matrices according to each theme to facilitate synthesis and interpretation.

### 2.4. Ethical Approval

The study was approved by the University of Nottingham’s Medical School Ethics Committee (D15092015).

## 3. Results

### 3.1. Sample Characteristics and Main Themes

The 22 participants included 14 males and eight females, who were predominantly white British, aged under 40 years and in employment (Table 1). Most (16) were exclusive e-cigarette users; four were dual users and one had returned to smoking after using e-cigarettes. Most (15) had been vaping for over a year. We identified six themes and corresponding sub-themes, which are listed in Table 2. The themes and sub-themes are summarised in Section 3.2, Section 3.3, Section 3.4, Section 3.5, Section 3.6 and Section 3.7, together with supporting quotes (attributed to participants according to sex, age group and smoking and vaping behaviour (Dual use—D; Vaping only—VO; PS—Past smoker).

### 3.2. Reasons for Using E-Cigarettes

#### 3.2.1. Quitting Smoking

Most participants identified e-cigarettes as a quitting tool, ‘*It’s just purely an aid to stop smoking*’ (F 60+ VO/PS). For some participants, e-cigarettes provided a gradual and staged approach to quitting smoking, in which participants reduced their nicotine intake to address the addiction prior to changing their smoking habits, ‘*It took some time to get there…being able to adjust the strength of my fluids …was helpful*’ (M26-30 D). Some seemed to have quit smoking tobacco cigarettes with relative ease, ‘*I quit straightaway…my fags were in the bin*’ (F 18-25 VO/PS), and reported reducing nicotine ‘*...happens naturally*’ (F 26-30 VO/PS).

Respondents who had quit were able to begin to feel that they had achieved something in terms of taking control of their smoking habit, ‘*I’ve done really well by not having the normal cigarettes*’ (M 31-39 VO/PS).

Most vapers who were using e-cigarettes to quit expressed a desire to eventually stop using them, ‘*my ultimate goal is to pack it in once and for all*’ (M 31-39 VO/PS), though some highlighted the difficulty of doing so: ‘*it’s breaking the physical habit of actually picking it up now that I have trouble with and that will come with time…It’s mostly just reminding yourself because there is no nicotine craving any more, it’s literally just a habit, same as biting my nails.*’ (F 26-30 VO/PS)

#### 3.2.2. Cutting Down Smoking

All dual users reported having cut down significantly on regular cigarettes, ‘*I used to get through a packet of 20 in about two days, if that sometimes. Now a pack of 20 will last me nearly two weeks*’ (F 18-25 D), and that using an e-cigarette had helped them to ‘*to try and cut down*’ (F 60+ VO/PS) or reduce their nicotine intake. Combustible cigarettes were more likely to be smoked when individuals were stressed, when ‘*it’s a bad day*’ (M26-30 D) or ‘*when I’ve had a drink…(I) would prefer to have a normal cigarette rather than vape*’ (M 18-25 D).

### 3.3. Experiences of E-Cigarette Products and Use

#### 3.3.1. Contemporary

E-cigarettes were considered to be a contemporary and modern form of smoking, with respondents describing them as ‘*all new and exciting*’ (M 31-39 D) and ever evolving, ‘*…there’s always something going on*’ (P12 M 26-30 VO/PS). E-cigarettes were also perceived as being ‘*technical*’ and as such required ‘*a slight level of knowledge*’ (M 31-39 VO/PS). This also added to the appeal for those who ‘*like gadgets*’ (F 26-30 VO/PS).

#### 3.3.2. Enjoyable

Most participants perceived e-cigarettes as offering an enjoyable experience, comparable to other leisure activities and hobbies, ‘*It’s like video games, I enjoy doing it*’ (M26-30 D). The variety of flavours and the types of kit were particularly appealing characteristics. For several of the respondents, e-cigarette use was perceived as ‘*not a way of fighting that addiction… but instead it’s something to have fun with*’ (P1 M 18-25 VO/PS). Consequently, for some of those who enjoyed e-cigarettes, there appeared to be little motivation to stop using them.

#### 3.3.3. Personal and Domestic Hygiene

E-cigarettes were often referred to by respondents as a ‘*cleaner*’ (P6 M 31-39 VO/PS) smoking experience. E-cigarettes were regarded as having a range of positive effects from making ‘…*the room smell really nice*’ (P10 M26-30 D) to improving personal hygiene, ‘*Your body doesn’t smell, your fingers don’t go yellow, the actual cloud you breathe out doesn’t cling to stuff*’ (P9 F 18-25 D). Unlike the smell of tobacco smoke, ‘*You walk past a vaper and you might get cherries or cola or menthol*’ (P12 M 26-30 VO/PS).

#### 3.3.4. Cost, Product Quality and Safety

Almost all interviewees referred to the lower cost of using e-cigarettes compared with cigarettes, with most respondents admitting ‘*saving a hell of a lot of money*’ (P11 M 31-39 VO/PS), ‘*I was spending about £1600 before I started vaping…now I would say about £11′* (P11 M 31-39 VO/PS). A minority mentioned that costs could escalate, ‘*You could spend a couple of hundred quid easily…buying loads of different liquids and buying loads of different gear*’ (P11 M 31-39 VO/PS). Cost was also an issue in terms of quality control with many respondents expressing that in terms of e-cigarettes, ‘*you get what you pay for*’ (P23 M 29). E-cigarette shops were considered to be specialist and often reputable suppliers, as such they adhered to certain standards, ‘*The shop we go to only stocks CE certified hardware and liquid.*’ Similar comments were made in relation to e-cigarette liquids, with a few expressing that ‘*you’ve got to be careful what liquids you do buy*’ (P18 F 50-59 VO/PS).

### 3.4. E-Cigarette Community and Culture

#### 3.4.1. Peer Introduction and Support

Family members, friends and partners were often instrumental in encouraging individuals to move from smoking cigarettes to using e-cigarettes, ‘*My new partner at that time suggested that…I try vaping. So that’s what I did*’ (M 31-39 VO/PS), where some participants reported trying their friends’ devices, ‘*My friend said, “Why don’t you try one of these pen things?”, so I had a go on hers, so I went out and bought one*’ (F 50-59 VO/PS).

#### 3.4.2. The Vape Shop Community

E-cigarette shops were considered by many respondents to be places that could provide accurate, reliable and current information about e-cigarette use, based on ‘*peers, other shop owners, the internet forums… and just personal knowledge*’ (F 26-30 VO/PS). The first visit to an e-cigarette shop was often lengthy and went beyond a simple purchase or transaction, often involving the demonstration of the different aspects of e-cigarettes by shop staff, ‘*We spent a bit of time with the shop lady and she showed me the different flavours, the different strengths you can get, she showed me the techniques on using it*’ (M 26-30 VO/PS).

Many respondents identified that ‘*you sort of become friends*’ (F 18-25 D). Most respondents reported feeling like they had become part of a community upon entering the shop and getting to know the local staff who they identified as being supportive and trustworthy, ‘*I felt welcomed in like they’re a little family in there, I felt like I was joining a team or something*’ (F 18-25 VO/PS). This was especially important for those using e-cigarettes as a quitting tool, who often felt vulnerable, ‘*You feel kind of small because you’re like, “I want to quit smoking. I don’t know how to do it. Help me.” You’re giving that trust to someone behind the counter*’ (M 26-30 VO/PS). Vape shops were also considered a source of guidance that was lacking from official sources, with several respondents expressing that e-cigarette use ‘*doesn’t seem to be regulated*’ (P20 M 26-30 VO/PS).

However, a minority of respondents expressed a differing perspective in which some felt ‘*All they wanted was a sale… they didn’t give me much advice*’ (M 60+ VO/PS).

#### 3.4.3. Community of E-Cigarette Users

Most participants acknowledged that they ‘*…like the sense of community within the e-cig group*’ (M 31-39 VO/PS). Vapers referred to being able to build ‘*connections*’ through sharing their own experiences through face-to-face encounters and via various online groups and forums, which were identified as a valuable way to ‘*find out what’s up and coming in the vaping industry*’ (M 26-30 VO/PS).

### 3.5. Benefits of E-Cigarettes Related to Level, Pattern and Location of Use

#### 3.5.1. Control over Smoking/Vaping Behaviour

E-cigarettes afford users a sense of freedom in terms of how devices can be used and according to their preferences. For example, respondents described how, with an e-cigarette, they can vape and then ‘*put it back down straight away and carry on with something else*’ (M 31-39 VO/PS). Others reduced the nicotine concentration in their e-liquid, ‘*it was too strong, so I dropped it down*’ (F 50-59 VO/PS). In comparison to tobacco cigarettes ‘*there’s alternatives with e-cigarettes*’ (M 31-39 VO/PS), with a few respondents reporting making their own liquids so that they could control what they were vaping, ‘*I’ve been making my own stuff and I know exactly what’s going in there*’ (M 31-39 VO/PS).

#### 3.5.2. Maintenance of Actions and Routines Associated with Tobacco Smoking

E-cigarettes were identified by respondents as a good substitute for smoking compared to other methods such as nicotine gum and patches because, with e-cigarettes, ‘…*you still go through the motions of having a cigarette*’ (M 31-39 D). The process of ‘*…inhaling and getting the hit*’ was visible with e-cigarettes whereas ‘…*with the patches, with the gum, it’s not the same*’ (M 31-39 D).

E-cigarettes allowed the user to maintain the same routines that had been built up around smoking. For example, although they could often smoke e-cigarettes inside, for some ‘*going outside*’ helped in ‘*fulfilling your rituals of smoking*’ (M 26-30 VO/PS) and, ‘*going outside and having an e-cigarette feels like having a cigarette*’ (M26-30 D). Furthermore, e-cigarettes enabled continued involvement in social groups such as joining friends or colleagues that still smoked, ‘*when the lads are having a break at work…I can go and do it with them and we can have a little chat*’ (M 60+ VO/PS).

#### 3.5.3. Use of E-Cigarettes in a Wide Range of Locations

E-cigarettes were identified by almost all respondents as being more socially acceptable than tobacco cigarettes, and as such could be used in any situation, ‘*I think you could go into someone’s house as a guest and say, “Is it okay if I smoke my e-cigarette?”, and everything will be fine with everyone*’ (M26-30 D). Some respondents’ perceptions that ‘*you can use it almost anywhere*’ (M 18-25 VO/PS) added to the belief that e-cigarettes were an easier and more acceptable smoking practice.

### 3.6. E-cigarettes and Health

#### 3.6.1. Short-Term Health Effects

Almost all respondents reported improvements in their health associated with using e-cigarettes. These included short-term improvements, ‘*My skin’s improved and my taste has improved*’ (M 31-39 VO/PS), with others highlighting that since using the e-cigarette they ‘*feel much better physically, mentally, just overall feel a lot better*’ (M 26-30 VO/PS) and that they ‘*…feel absolutely amazing now compared to how I did when I smoked*’ (M 18-25 VO/PS).

Some participants had noticed some negative health effects since they had begun smoking e-cigarettes, ‘…*sometimes I will feel a bit sick after having it*’ (F 18-25 VO/PS). Others noticed more general side effects, ‘…*it can dry out the back of your throat*’ (M26-30 D). Many respondents highlighted that e-cigarettes initially seemed harsher and more abrasive than tobacco cigarettes, ‘*it just absolutely choked me, hurt my throat and made me cough*’ (F 50-59 VO/PS), although there were differences between products, *“One made me cough, one didn’t”* (M 60+ VO/PS).

#### 3.6.2. Perceptions of Long-Term Health Effects

The prevailing view was that e-cigarette use is ‘…*not without its dangers*’ (M 29) but ‘*safer and a lot better for you*’ (M 18-25 VO/PS) than tobacco smoking. This was supported by awareness of the negative health effects of traditional cigarettes, ‘*There’s no links to say e-cigarettes will kill you, whereas actual cigarettes will kill you*’ (M 31-39 VO/PS). Despite the many positive statements given by respondents a few stated that ‘*If there was any health disadvantages to it [e-cigarettes] in the long run, even fairly moderate ones then I would give it up*’. (M26-30 D).

Despite an overarching view that e-cigarettes were less harmful than smoking, there were also numerous anxieties expressed around the ‘*unknowns to electronic cigarettes and vaping*’ (M 26-30 VO/PS), with several mentioning that ‘*there’s not much research on* e-*cigarettes*’ (M 31-39 VO/PS). While several respondents had attempted to gather information about e-cigarettes, ‘*I did get it up on the internet and I did read an awful lot about them*’ (F 50-59 VO/PS), the information available was considered to be contradictory and confusing, ‘*you’d read something that would say that it’s bad for you, then you’d read something to say they don’t know…I didn’t really know what to believe*’ (F 50-59 VO/PS). Some respondents identified health risks for which there is little evidence ‘*e-liquids can cause what they call a popcorn lung*’ (F 50-59 VO/PS) and expressed concerns which did not take account of the risk relative to combustible to tobacco use ‘*I looked up the stuff that’s in it and realised that it actually wasn’t safe at all*’ (M 26-30 VO/PS).

#### 3.6.3. Concerns about Overuse/addiction

Some participants were concerned about continuing addiction, stating that although e-cigarettes had helped them quit smoking they were, ‘*on my vape nearly all the time now*’ and as such felt that they were ‘*taking an addiction from one thing and getting an addiction to another*’ (M 18-25 VO/PS). Some felt concerned that e-cigarettes were not really a form of stopping smoking, ‘*I’m not effectively stopping myself from smoking…Because I feel like I’m doing it more*’ (M 18-25 D).

### 3.7. E-cigarettes and Cessation Support

#### 3.7.1. E-Cigarettes and Smoking Cessation Services

Many respondents felt that at present there was little support from traditional smoking cessation services in relation to e-cigarettes, with the perception that they ‘*don’t really deal with e-cigs*’ (F 26-30 VO/PS). Some respondents expressed an interest in quitting e-cigarettes but were ‘*not sure how to do it and how to tackle* it’ (F 50-59 VO/PS) or where to access advice and support regarding e-cigarettes.

#### 3.7.2. Smoking Cessation in Vape Shops

Vape shops were not identified by users as a place to receive smoking cessation support, ‘*they never actually said along the lines of “this will help you quit”*’ (F 26-30 VO/PS), although most users did not report having asked for quitting advice either. Despite this, e-cigarette shops were acknowledged by most respondents to be potentially suitable places in which to provide cessation support, ‘*If I’d have gone into that first shop and the bloke behind the counter was trained in quitting and cutting down, I think I could have got some really good advice there*’ (M 26-30 VO/PS). Having a personal trusting relationship with the provider of quitting advice was considered important for many respondents, who valued ‘*relatable, personal evidence*’ (M 26-30 VO/PS). Many respondents expressed that ‘…*they need to obviously promote [quitting] a bit more in the shops*’ with several admitting that they would ‘*…like a bit more advice on it really when I go into a shop*’ (F 18-25 VO/PS).

This view was not shared by all respondents, however, with others suggesting that they would not be best placed to provide quitting support because, ‘*They’re a business, they’re trying to sell their products*’ (M 31-39 VO/PS). Several respondents felt that most people would ‘*take advice elsewhere for that sort of thing*’ (F 50-59 VO/PS) rather than in e-cigarette shops.

## 4. Discussion

This study investigated the role of vape shops in vaping behaviour and the potential for vape shops to support smoking cessation from the perspective of vape shop customers. Our findings reveal that most respondents regarded e-cigarettes as a quitting tool, with several reporting having quit smoking as soon as they took up vaping; and that vape shops were central to their positive experiences of vaping in that they provided access to a wide variety of high-quality products and reliable product information and advice. Vape shops were identified as hubs of the vaping community which are staffed by trustworthy, knowledgeable individuals who take the time to give detailed product advice. The shop staff engendered a sense of loyalty in customers which, together with the community of other vapers, created a network which helped to support e-cigarette use. Vape shops were not, however, regarded as a setting in which to seek out cessation-specific advice, and participants reported not knowing where to go for this type of information. Vape shops were acknowledged as potentially appropriate places to provide quitting support, although some participants alluded to a potential conflict of interest vape shops in doing so.

Very little research has previously been undertaken in vape shops, and, in particular, very few studies have explicitly explored the role of vape shops in supporting smoking cessation. Our sample may be biased in favour of keen vapers and vapers who have quit smoking; however, we recruited customers directly within vape shops, which is likely to provide a more representative sample of vape shops customers than other recruitment methods such as social media. Our study was conducted in a single region of the UK, but its findings provide valuable insights for researchers and policy makers in other countries around the views and behaviours of vape shops customers and the role that vape shops play in supporting tobacco harm reduction.

Previous studies suggest that vape shop staff regard themselves as being able to educate customers about e-cigarette use and that access to staff is a key reason for people to use vape shops [15,19]. E-cigarette products vary widely and require a level of technical knowledge and, as such, there is a ‘vaping learning curve’ to their use [20]. Vape shops are therefore likely to be integral to supporting smoking cessation using e-cigarettes, as they can ensure that customers choose products that are most suitable to them and use them correctly—a process which is generally lacking in randomized trials of e-cigarettes for smoking cessation and where the type of device tested is likely to influence quit rates [21]. Previous research has demonstrated that vape shops play an important role in providing product advice to smokers who are new to vaping, but also ongoing support to existing vapers [7]. Furthermore, vape shop staff may provide product repairs and advice to smokers who have relapsed [7]. These examples emphasize that short-term guidance alone may not be sufficient to maximise the role that vaping and vape shops can play in maintaining smoking abstinence.

Our findings are in line with existing studies which indicate that many vapers regard vaping as an enjoyable activity [20,22,23,24,25], with the variety of available products contributing to its appeal. Given the limited range of vaping products available in other retailers (such as supermarkets), this further highlights the extent to which vape shops may be important in maintaining interest in vaping and supporting smoking abstinence.

While vape shops could have an important role to play in smoking cessation, this study echoes the findings of previous studies which found that smoking cessation is not a core aspect of the dialogue within vape shops [7,13]. As such, some opportunities to support quit attempts among vape shop customers may be being missed. Many participants felt that inadequate support for using e-cigarettes in quit attempts was available, and that receiving cessation advice in vape shops, rather than product advice alone, would be useful. Some level of co-working between smoking cessation services and vape shops has been suggested previously [7], and a model whereby shop staff are trained to deliver cessation advice may be most well-received [16]. Previous studies have identified vapers who are resistant to the vaping culture and community [7,23], and of whom some perceive vaping as a medical treatment rather than an enjoyable or sociable activity. It is also possible that some smokers are deterred from experimenting with e-cigarettes by the perceived vaping culture and might find an offer of more formal cessation advice, without forgoing the benefits of in-depth product advice, appealing. Any vape shop-based cessation intervention would, however, have to strike a balance between encouraging e-cigarette uptake in smokers who do not currently vape, and not deterring existing customers who may regard vape shops as an inappropriate setting for providing smoking cessation advice.

Many participants in our sample were happy to continue vaping in the future; to them, switching completely from tobacco to e-cigarette constituted ‘quitting’. However, others were concerned about continuing addiction and expressed a desire to give up vaping. While helping smokers to quit tobacco cigarettes is a clear benefit of e-cigarettes, these products are likely to be associated with some health risks (albeit that existing evidence indicates that these risks are likely to represent a very small fraction of the risks posed by smoking [2]), and their use also imposes a financial burden. It is therefore important to develop services or strategies to help established vapers to quit vaping. Our participants drew many parallels between vaping and cigarette smoking, suggesting that some of the same behavioural cues that would need to be addressed would be similar.

While the vape shops in our study provided a significant amount of product information, and participants generally reported improvements in their health, several expressed uncertainties around the harms of e-cigarettes. Misperceptions about the harms of e-cigarettes are widespread [2], and our findings illustrate that there is confusion in people who are currently vaping, not just in the general population. For example, in our sample, some participants expressed serious concerns about the health effects of vaping which do not reflect the existing evidence base. This underlines the importance of providing clear and reliable information on e-cigarettes. In particular, this requires that any information about the potential harms of e-cigarettes is presented in comparison to the risks posed by combustible tobacco use.

Overall, therefore, our study suggests that vape shops provide a valuable resource for smokers trying to cut down or quit tobacco use, but also represent an opportunity to deliver more extensive services aimed at long-term cessation of nicotine use as well as long-term substitution of tobacco with electronic cigarettes. Vape shops also offer a medium through which to deliver independent information on the relative risks of tobacco and e-cigarette use. Our study also indicates that vape shops could do more to appeal to smokers who have yet to try e-cigarettes, possibly by offering more formal pathways to quitting. Although doing so presents an apparent conflict of interest, our findings suggest that such an approach is most likely to interest individuals who are not currently vaping. Since the prevalence of e-cigarette use now appears to have plateaued in England [1], this represents an opportunity to attract new customers and hence new business. A further commercial opportunity lies in marketing existing and new services to low income smokers, who as a group have in the past been slower to adopt electronic cigarettes than higher income smokers [26].

## 5. Conclusions

Our study suggests that vape shops have the potential to play an important role in tobacco harm reduction, particularly if their service model could extend to help smokers to quit. Research efforts should focus on investigating how to maximize those opportunities in the vape shop setting.

## Figures and Tables

**Table 1 ijerph-16-02341-t001:** Participant characteristics.

Question	*n* (%)
Sex	
Male	14 (64)
Female	8 (36)
Age	
18–25	5 (23)
26–30	6 (27)
31–39	4 (18)
40–49	0 (0)
50–59	3 (14)
60+	4 (18)
Ethnic group	
White	20 (91)
Asian/Asian British	1 (4)
Unknown	1 (4)
Education	
Degree level or equivalent	3 (14)
A-Level, equivalent or higher education below degree level *	7 (32)
GCSE or equivalent *	5 (23)
Other below degree level	1 (4)
No formal qualifications	5 (23)
Unknown	1 (4)
Employment	
In full-time work	12(55)
In part-time work	1 (4)
Retired	3 (14)
Student	2 (9)
Not in work (long-term sick, disabled, unemployed, other reason)	4 (18)
Smoking/vaping behaviour	
E-cigarettes only and previously smoked	16 (73)
Dual user (e-cigarettes and cigarettes)	4 (18)
Cigarettes only	1 (4)
Unknown	1 (4)
Duration of e-cigarette use	
1–4 weeks	1(4)
6 months–1 year	5 (23)
1–2 years	11 (50)
2 years +	4 (18)
Unknown	1 (4)
Current average nicotine concentration used	
0	3 (14)
1–3	6 (27)
4–10	5 (23)
11–20	7 (32)
No response	1 (4)

Note: Percentages are rounded to the nearest whole number. * General Certificates of Secondary Education (GCSEs) and Advanced levels (A-levels) are secondary school leaving qualifications in the UK. GCSEs are typically taken at age 16, A-levels at age 18.

**Table 2 ijerph-16-02341-t002:** Themes and sub-themes.

Themes	Reasons for Using E-Cigarettes	Experiences of E-Cigarette Products and Use	Community and Culture Surrounding E-Cigarettes	Benefits of E-Cigarettes Related to Level, Pattern and Location of Use	E-Cigarettes and Health	E-Cigarettes and Cessation Support
Sub-themes	Quitting smoking Cutting down on smoking	Contemporary Enjoyable Personalisation Cost, product quality and safety	Peer introduction and support The vape shop community Community of e-cigarette users	Control over smoking/vaping behaviour Maintenance of actions and routines associated with tobacco smoking Use of e-cigarettes in a wide range of (indoor) locations	Acute health effects Long-term health effects Concerns about overuse/addiction	E-cigarettes and smoking cessation services Smoking cessation in vape shops

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
