# Peer review of "‘I Felt Welcomed in Like They’re a Little Family in There, I Felt Like I Was Joining a Team or Something’: Vape Shop Customers’ Experiences of E-Cigarette Use, Vape Shops and the Vaping Community"

_ijerph, 2019, doi:10.3390/ijerph16132341_

Round 1

Reviewer 1 Report

Fantastic paper, and refreshing to read about vape shops not in the US!

Regarding line 324: "these products are unlikely to be harmless in the long term," maybe it would be helpful to phrase this another way, especially since there is current research out there pointing to harmful effects of e-cigarettes, like how you mentioned in lines 334-35: "In particular, this requires that any information about the potential harms of e-cigarettes is presented in comparison to the risks posed by combustible tobacco use."

Also in the US, the FDA is actually against vape shop employees giving out advice (due to potential miscommunication, etc), so in the UK is there anything similar where a government agency may be against employees giving cessation advice? Just wondering :)

Author Response

Fantastic paper, and refreshing to read about vape shops not in the US!

Response: Many thanks to the reviewer for their positive review.

Regarding line 324: "these products are unlikely to be harmless in the long term," maybe it would be helpful to phrase this another way, especially since there is current research out there pointing to harmful effects of e-cigarettes, like how you mentioned in lines 334-35: "In particular, this requires that any information about the potential harms of e-cigarettes is presented in comparison to the risks posed by combustible tobacco use."

Response: We have changed this sentence, which now reads:

“…these products are likely to be associated with some health risks (albeit that existing evidence indicates that these risks are likely to represent a very small fraction of the risks posed by smoking)…”

Also in the US, the FDA is actually against vape shop employees giving out advice (due to potential miscommunication, etc), so in the UK is there anything similar where a government agency may be against employees giving cessation advice? Just wondering :)

Response: Our understanding is that the reviewer is not asking for us to add this information to the manuscript but is seeking clarification. In the UK there has been no guidance to this effect.

Reviewer 2 Report

This is an interesting paper which adds to the growing evidence base on the role of vape shops in supporting its customers. It is very well written and I only have a few minor suggestions.

The first is that the authors should develop the role of vape shops for both 1) smokers, and 2) vapers who have transitioned. The distinction between the two needs greater clarity. Vape shops are providing support in two ways and are able to adapt by to the transitioning vaper. Vape shops also provide support for those who have lapsed - the detail of this can be found in the Ward et al., paper that the authors cite (7). 

The Ward et al paper was not conducted in only East Anglia, three of the vape shops were in Kent and London - this needs to be changed in the introduction.

The themes could benefit from more examples and highlight any that conflicted with the general consensus of thinking or where themes highlighted flaws in regulation. 

Enjoyment and Pleasure - this is really important for the transitioning and transitioned vaper. From a behaviour change perspective, motivation for the target behaviour (cessation) must outweigh competing factors, so the role of vaping in keeping and maintaining interest should be given more weight in the discussion. 

See this paper on DIY home-mixing for example - Cox, S., Leigh, N. J., Vanderbush, T. S., Choo, E., Goniewicz, M. L., & Dawkins, L. (2018). An exploration into “do-it-yourself”(DIY) e-liquid mixing: Users' motivations, practices and product laboratory analysis. Addictive Behaviors Reports, 100151. 

Pleasure and enjoyment are shown to keep vapers interested and importantly the products play a role in maintaining interest away from smoking. Similarly, Notley, has provided examples of vapers taking on new identities, and that that these have been founded or reinforced by vaping communities - often started within vape shops. 

Notley, C., Ward, E., Dawkins, L., & Holland, R. (2018). The unique contribution of e-cigarettes for tobacco harm reduction in supporting smoking relapse prevention. Harm reduction journal15(1), 31.

Notley, C., Ward, E., Dawkins, L., Holland, R., & Jakes, S. (2019). Vaping as an alternative to smoking relapse following brief lapse. Drug and alcohol review38(1), 68-75.

Overall, a good paper with only minor edits and greater discussion around the varying role of vape shops on quitting trajectories needed. 

Author Response

This is an interesting paper which adds to the growing evidence base on the role of vape shops in supporting its customers. It is very well written and I only have a few minor suggestions.

Response: Many thanks to the reviewer for their positive review.

The first is that the authors should develop the role of vape shops for both 1) smokers, and 2) vapers who have transitioned. The distinction between the two needs greater clarity. Vape shops are providing support in two ways and are able to adapt by to the transitioning vaper. Vape shops also provide support for those who have lapsed - the detail of this can be found in the Ward et al., paper that the authors cite (7). 

Response: We have made a small change to the introduction to reflect this and have added a related paragraph to the discussion:

“Their customers are likely to be interested in quitting smoking or cutting down, and staff interactions with customers can provide opportunities to promote and maintain smoking cessation.[7]”

“Previous research has demonstrated that vape shops play an important role in providing product advice to smokers who are new to vaping, but also ongoing support to existing vapers. [7] Furthermore, vape shop staff may provide product repairs and advice to smokers who have relapsed. [7] These examples emphasize that short term guidance alone may not be sufficient to maximise the role that vaping and vape shops can play in maintaining smoking abstinence.”

The Ward et al paper was not conducted in only East Anglia, three of the vape shops were in Kent and London - this needs to be changed in the introduction.

Response: We have corrected this in the introduction.

The themes could benefit from more examples and highlight any that conflicted with the general consensus of thinking or where themes highlighted flaws in regulation. 

We have added some examples of concerns about the health risks of vaping that were expressed by participants which are not in line with the existing evidence base:

“Some respondents identified health risks for which there is little evidence ‘e-liquids can cause what they call a popcorn lung’ (F 50-59 VO/PS) and expressed concerns which did not take account of the risk relative to combustible to tobacco use ‘I looked up the stuff that’s in it and realised that it actually wasn’t safe at all’ (M 26-30 VO/PS).”

We have added a sentence to the discussion which relates to this:

“For example, in our sample some participants expressed serious concerns about the health effects of vaping which do not reflect the existing evidence.”

Furthermore, we have added an example related to perceptions of inadequate information and regulation from official sources: 

“Vape shops were also considered a source of guidance that was lacking from official sources, with several respondents expressing that e-cigarette use ‘doesn’t seem to be regulated’ (P20 M 26-30 VO/PS). “ 

Enjoyment and Pleasure - this is really important for the transitioning and transitioned vaper. From a behaviour change perspective, motivation for the target behaviour (cessation) must outweigh competing factors, so the role of vaping in keeping and maintaining interest should be given more weight in the discussion. 

Pleasure and enjoyment are shown to keep vapers interested and importantly the products play a role in maintaining interest away from smoking. Similarly, Notley, has provided examples of vapers taking on new identities, and that that these have been founded or reinforced by vaping communities - often started within vape shops. 

Response: We have added the following paragraph to the discussion with additional references:

"Our findings are in line with existing studies which indicate that many vapers regard vaping as an enjoyable activity [20, 22-25], with the variety of available products contributing to its appeal. Given the limited range of vaping products available in other retailers (such as supermarkets), this further highlights the extent to which vape shops may be important in maintaining interest in vaping and supporting smoking abstinence."